# Pain Control Affects the Radiographic Diagnosis of Segmental Instability in Patients with Degenerative Lumbar Spondylolisthesis

**DOI:** 10.3390/jcm10173984

**Published:** 2021-09-02

**Authors:** Shih-Hsiang Chou, Sung-Yen Lin, Po-Chih Shen, Hung-Pin Tu, Hsuan-Ti Huang, Chia-Lung Shih, Cheng-Chang Lu

**Affiliations:** 1Department of Orthopaedics, Kaohsiung Medical University Hospital, Kaohsiung Medical University, Kaohsiung 807, Taiwan; stanelychou@gmail.com (S.-H.C.); tony8501031@gmail.com (S.-Y.L.); shenporch@gmail.com (P.-C.S.); hthuangtka@gmail.com (H.-T.H.); 2Orthopaedic Research Centre, Kaohsiung Medical University, Kaohsiung 807, Taiwan; 3Graduate Institute of Medicine, College of Medicine, Kaohsiung Medical University, Kaohsiung 807, Taiwan; 4Department of Orthopaedics, College of Medicine, Kaohsiung Medical University, Kaohsiung 807, Taiwan; 5Department of Public Health and Environmental Medicine, School of Medicine, College of Medicine, Kaohsiung Medical University, Kaohsiung 807, Taiwan; p915013@kmu.edu.tw; 6Clinical Medicine Research Center, Ditmanson Medical Foundation Chia-Yi Christian Hospital, Chia-Yi City 600, Taiwan; stone770116@gmail.com; 7Department of Orthopaedics, Kaohsiung Municipal Siaogang Hospital, Kaohsiung Medical University, Kaohsiung 807, Taiwan

**Keywords:** segmental instability, flexion and extension radiography, ketorolac, low back pain, spondylolisthesis

## Abstract

Background: Diagnosing intervertebral instability is crucial for the treatment of degenerative lumbar spondylolisthesis (DLS). Disabling back pain will reduce spinal mobility which leads to an underestimate of the incidence of intervertebral instability. We hypothesized that adequate analgesia could alter the flexion/extension exam performance, and thus increase the diagnostic accuracy of segmental instability. Materials and methods: One hundred patients with low-grade DLS were prospectively enrolled in the before–after cohort study. Standing lateral flexion/extension radiographs of lumbar spines were examined and analyzed before and after intramuscular injections of 30 mg ketorolac. Results: Pain score decreased significantly after analgesic injections (*p* < 0.001). Dynamic slip (DS), dynamic segmental angle (DA), dynamic lumbar lordosis, and slip percentage (SP) were significantly increased after pain reduction (all *p* < 0.001). According to the diagnostic criteria for segmental instability (DS > 4.5 mm, DA > 15°, or SP > 15%), there were 4%, 4%, and 0.7% of total motion segments fulfilling the criteria which markedly increased to 42%, 32%, and 16.7% after analgesia was administered. The incidence of instability also increased from 6% to 38% after analgesia. Conclusions: The diagnosis rate of intervertebral instability is commonly underestimated in the presence of low back pain. This short-term pain relief facilitates reliable functional imaging adding to the diagnosis of intervertebral instability.

## 1. Introduction

Low back pain (LBP) is the most common musculoskeletal disorder and the leading cause of disability endured for years [1]. About 20% of patients affected by acute LBP may develop chronic LBP in one year [2]. A recent study further demonstrated that 32% of patients in the United States with acute LBP seeking primary care transited to chronic LBP at 6 months [3]. Spondylolisthesis with segmental instability can lead to disabling LBP symptoms and neurologic deficits [4], and the management tends to involve spinal fusion surgery when conservative treatment has failed [5,6]. Degenerative lumbar spondylolisthesis (DLS) is a common cause of chronic LBP, and the evaluation of segmental stability is considered an important factor to determine the treatment of patients with DLS [5,7]. Although there is no consensus about the definition of segmental instability, a spinal motion segment is often considered mechanically unstable when it exhibits increased or abnormal motion; therefore, the measurement of lumbar spine dynamics is fundamental [8,9].

The decision to perform spinal fusion requires evidence of intervertebral instability, but it is difficult to define the segmental instability. Several methods including the detection of facet joint effusion in T2-weighted magnetic resonance imaging [10]; the measurement of facet joint volume, as called “facet opening”, in three-dimensional reconstruction computed tomography [11]; or intraoperative measurements [5] were recently proposed to identify segmental instability. Nevertheless, flexion/extension functional radiographs are considered the gold standard method for diagnosing DLS [12] and quantifying segmental instability [13]. This method is often combined with clinical findings to determine whether spinal fusion is necessary [14,15].

Pain is an important factor affecting spinal activity, but the reported results of pain reduction on motion improvement are controversial. It was reported that the spine motion arc and curvature in patients with chronic LBP could be restored after pain control [16]. On the contrary, some authors suggested that simply targeted pain relief had little or no effect on the range of motion [17,18]. However, previous studies did not report the results based on actual flexion/extension radiographs, and neither focused on the motion segment movements. Little is known about the effects of pain relief on lumbar instability interpretations. In the present study, we hypothesized that reduced pain could alter the flexion/extension exam performance, thus affecting the diagnostic accuracy of segmental instability. The purpose of this study is to investigate the pain control efficacy of fast-acting analgesia in patients with chronic symptomatic DLS and the changes in radiologic parameters of intervertebral instability in lateral flexion/extension radiographs of the lumbar spine.

## 2. Materials and Methods

### 2.1. Patient Recruitment

This before–after cohort study was approved by our institutional review board (KMUH-IRB-F(I)-20170129). Informed consent was obtained from each patient before examinations. From February 2018 through December 2019, patients with the presenting chief complaint of mechanical LBP defined as worsening symptoms with standing and sitting for periods or upon standing from the seated position, or bending forward, with/without radicular symptoms were recruited. The inclusion criteria were as follows: patients with LBP lasting 6 months or more and diagnosed with low grades (Meyerding grade I and II) [19] of DLS by standard lumbar X-rays, including anteroposterior and lateral radiographs. Patients were screened for exclusion based on their clinical history and blood examinations. The exclusion criteria were as follows: age <20 years or >70 years, history of spine surgery, congenital deformity, current neurological disorder, allergy to non-steroidal anti-inflammatory drugs (NSAIDs), pregnancy, and moderate-to-severe kidney failure (≥stage 3).

### 2.2. Analgesia Drug Injection and Lumbar Flexion and Extension Radiography

Intramuscular (IM) injection with ketorolac 30 mg (30 mg/L Amp) (YUNG SHIH PHARM. IND. CO., Ltd., Taichung, Taiwan) was adopted for the treatment of back pain. Ketorolac is a NSAID that is commonly used for the short-term treatment of moderate to severe pain after a medical procedure or after surgery. The basic vital signs including blood pressure, heart rate, temperature, and respiration rate were recorded within the 30–40 min after injection, and patients with abnormal vital signs were excluded from the study. Side effects after injection were recorded, such as nausea/vomiting, headache, or dizziness. Standing lateral lumbar flexion/extension radiographs were taken before and after 30-40 min of ketorolac injection depending on the condition of preparation. The time interval between ketorolac injection to radiography was determined according to the pharmacokinetics [20] and the pharmaceutical company’s guidelines. Images were acquired on 10/17 inch digital X-ray cassettes with a film focus distance of 100 cm and at 90 KV. Patients were asked to flex or extend their backs as much as possible during examinations.

### 2.3. Clinical Evaluation

The visual analog scale (VAS) scoring system was used to evaluate back pain level in patients before and 30 min after analgesia. We recorded baseline and final VAS scores with reference to radiographic performance. The difference in VAS scores between pre- and post-analgesia was recorded.

### 2.4. Radiographic Evaluation

The digital images were evaluated on a Picture Archiving and Communication System (PACS). Sagittal translation and segmental angulation on lateral flexion/extension radiographs were used to assess segmental instability and the results were measured by two observers who were completely blinded to all information, including the age, name of patients, and time of image. The mean measurement values between the two observers were adopted for analysis. Target motion segment (TS) comprised two vertebrae; forward or backward displacement of one vertebra over a lower vertebra detected by previous standard lumbar X-rays was named according to the position of the two vertebrae. Radiographic measurement parameters were as follows: (1) Vertebral body width (mm); (2) Segmental translation (mm); (3) Segmental angulation (°); (4) Lumbar lordosis angle (LA) (°); (5) Dynamic slip (DS) (mm); (6) Dynamic segmental angle (DA) (°); (7) Dynamic lumbar lordosis (DL) (°); and (8) Slip percentage (SP) (%), defined in Figure 1. Segmental instability was defined as SP > 15%, DS > 4.5 mm, DA > 15° (L1/L2, L2/L3, or L3/L4), DA > 20° (L4/L5), or DA > 25° (L5/S1) [8]. Except for the previously identified TS, additionally recognized motion segments with spondylolisthesis after analgesia were added for analysis. According to baseline median VAS score, patients were divided into moderate pain and severe pain groups, and the radiographic measurements were compared between the two groups.

### 2.5. Statistical Analysis

All analyses were performed with SPSS version 19.0 (SPSS Inc., Chicago, IL, USA). Continuous variables were presented as mean ± standard deviation. The intraclass correlation coefficient (ICC) two-way random model of absolute agreement was used to analyze measurement reliability. A paired-*t* test was used to assess the difference between pre- and post-injection. Pre–post changes within different groups were estimated via the standardized response mean, with mean differences between pre- and post-analgesia divided by the standard deviation of the difference parameters. Adjusted mean differences (AMDs) were calculated to quantify the between-group effects. General linear models were used to assess the difference between the two sample groups. Linear regression analysis was used to evaluate the relationship between radiograph measurement and pain scores. *p* < 0.05 was considered significant.

## 3. Results

### 3.1. Baseline Cohort Analyses

Demographic data and radiographic measurements of the 100 patients enrolled in the study are shown in Table 1. A total of 150 TSs were observed: one level was observed in 56 patients, two levels in 38 patients, and three levels in 6 patients. Of the 150 TSs, there were 7 TSs located at L2/L3, 48 at L3/L4, 88 at L4/L5, and 7 at L5/S1.

### 3.2. The Changes in Pain Intensity and Radiographic Parameters after Analgesia

Generally, ketorolac IM injections provided significant analgesia for LBP. After injection, the VAS score was significantly dropped down (34.7 ± 15.5 mm, range 10–70 mm; *p* < 0.001, Table 1). The average segmental translation and angulation in flexion and extension were both significantly increased after analgesia, resulting in a significant increase in DS and DA (*p* < 0.0001 in both). The mean DL of post-analgesia (42.61 ± 12.94) was 109.17% compared with 39.03 ± 13.61 pre-analgesia. The mean DL was markedly increased by 9.17% after analgesia, and the SP significantly increased after injection (mean difference = 6.23%, *p* < 0.001). The measurement reliability of radiographic variables was high; the ICC ranged from 0.958 to 0.997, and SEM ranged from 0.012479 to 0.099168. The measured results in lateral flexion/extension radiographs before and after analgesic injection are presented in Table 2 and a representative case is demonstrated in Figure 2.

### 3.3. The Detection of Instability Segments after Analgesia Relative to Prior Analgesia

When DS > 4.5 mm was considered as instability, 63 of 150 (42%) total TSs showed instability after analgesia compared with those without analgesia (6 of 150 TSs; 4%). On the other hand, 48 of 150 total TSs (32%) exhibited instability after analgesia compared with those without treatment (6 of 150 TSs; 4%) while using the segmental angulation criteria to determine instability. In considering the SP instability as criteria, the patients after analgesia also exhibited more instability (25 of 150 TSs; 16.7%) compared with those before analgesia (1 of 150 TSs; 0.7%). After injection, the number of patients diagnosed with unstable spondylolisthesis (segmental movements reached the diagnostic criteria of intervertebral instability) was increased from 6 (before analgesia) to 38 (after analgesia) (Figure 3).

### 3.4. The Difference in Motion Change in Moderate and Severe Pain Subgroups

The patients were divided into moderate pain (46 patients, average VAS: 58 mm) and severe pain groups (54 patients, average VAS: 75 mm) based on the median baseline VAS scores (70 mm) (Table 3). We observed a significantly lower extension angle in segmental angulation pre-analgesia in the severe pain group (8.92 ± 4.08) compared with the moderate pain group (10.27 ± 4.07) (*p* = 0.046). In regard to the radiographs taken after the injection, the severe pain group had a significantly greater increase in DA than the moderate pain group (*p* = 0.03). Although there was no significant extension and flexion angle increase between the two groups, the increase in extension angle after the intervention was approaching significance (*p* = 0.07). Reevaluation tests demonstrated that the results were similar and the multiple testing problem was not significant.

### 3.5. Association between Pain Score and Radiographic Parameters

The averaged DS and DA exhibited a negative correlation with pain scores. The plot of DS versus VAS score (including baseline and final VAS scores) showed an inverse relationship. The regression equation was y = 5.73−0.52x (R^2^ = 0.335, *p* < 0.001), where y is the DS and x is the VAS score. Similarity, DA and pain scores also showed an inverse relationship (equation: y = 18.71−1.09x, R^2^ = 0.199, *p* < 0.001) (Figure 4).

## 4. Discussion

The present study examined the effect of alleviating LBP on the variables measured from flexion/extension radiographs of the lumbar spine. We found that 30 mg of IM administered ketorolac was effective for alleviating LBP and caused an increased motion of spondylolisthesis segments in aspects of 2.4 (±1.36) mm in DS, 4.85 (±4.59) degrees in DA, and 6.23 (±3.46)% in SP. About 32% more patients with segmental instability were detected in lateral flexion/extension radiography after pain relief. Severe pain (VAS > 70 mm) could prevent patients from performing adequate flexion/extension exercises, resulting in an underestimation of the segmental instability.

The reoperation rate of DLS has been reported to be high both in either decompression alone or decompression plus fusion surgery [21,22,23,24]. Although widely investigated, the exact mechanisms of reoperation remain elusive. The instability occurrence on adjacent segments was the most common pathology in lumbar fusion or dynamic stabilization surgery [21,22,23]. Patients with low-grade DLS are more likely prone to experience postoperative instability after decompression laminectomy surgery in the presence of segmental translation > 1.25 mm, disc height > 6.5 mm, and facet angle > 50° [25]. The determination of instability in the index level of decompression surgery or adjacent levels of fusion surgery was particularly vital at the time of the primary surgery via either decompression alone or decompression with instrumental fusion. Our results indicate that the severity and incidence of segmental instability might be underestimated in patients suffering from LBP. To the best of our knowledge, the present study was the first to demonstrate segmental instability parameters after analgesia, and the results show that segmental translation and angulation values were significantly increased after analgesia. Patients could not fully flex or extend their backs due to pain. As a result, a substantial number of hidden spondylolisthesis motion segments would be detected after pain reduction. We believe that effectively detecting the underestimated instability masked by back pain could reduce the reoperation rate.

Pain prohibits normal spinal mobility [26] and muscle coordination [27]. Pain reduction contributes to normalizing ROMs in patients with chronic LBP [28]. Back pain generally reduces motion segment movement; however, pain reduction itself did not contribute equally to individuals in our study population. Patients with severe pain (VAS > 70 mm) benefited more from pain relief with a single dose of pain adjuvant than those with moderate pain; meanwhile, they experienced a marked increase in motion segment dynamic angles and flexion angles. Although this study lacked electromyography evidence, the above findings may be explained by the fact that flexion/relaxation was achieved after the pain reduction [29]. Additionally, the extension movements would be more limited in patients with severe LBP because of a weaker muscle strength in extension than in flexion [30].

Analgesic methods and outcomes of spinal mobility examinations from previous studies were inconsistent [16,17,18] (Appendix A). Moreover, there was no study using the motion changes in lateral flexion/extension radiographs to explore the influence of pain control on spinal mobility and on lumbar intervertebral instability. Lumbar facet joint injection or denervation has been indicated for diagnostic assessment or for pain reduction and mobility improvement in patients with painful facet joint syndrome or spondylolysis. However, lumbar facet joint injection requires the assistance of imaging guidance that is technically demanding and also not convenient for pre-operative evaluation [31]. Transcutaneous electrical nerve stimulation has been in use to treat LBP for several decades and is considered effective to reduce pain and improve spinal activity [16]. Nevertheless, this treatment requires several weeks, and may not provide a stable effect on pain reduction. Considering all treatment methods, injection with short-acting NSAID is obviously effective, convenient, less invasive, and may be more cost-effective than others for instability diagnosis. Accordingly, we argued that a single dose of ketorolac (30 mg) IM injection before radiological examinations was safe and efficient in alleviating pain for accurate instability detection.

There were several limitations to this study. First, we did not investigate whether pain reduction could decrease back muscle splinting. Further studies are necessary to shed more light on measuring flexion relaxation by using the electromyography method. Second, the leg tension sign was not classified and the effects of analgesia on the tension sign were not evaluated. Third, this study cohort lacked patients with mild pain and lacked a comparative group with placebo injections, and whether these patients would demonstrate similar results is uncertain. Finally, there is a substantial risk that an order effect of participant performance in repeated flexion and extension radiographs may have influenced results. Future studies should include the placebo injection and investigate whether repeated measurements might demonstrate increases in ROM even without analgesia, or whether the motion arc returns to pre-analgesia levels once the analgesia has worn off.

## 5. Conclusions

The diagnosis rate of intervertebral instability is underestimated in moderate to severe LBP. The intervertebral sagittal translation, segmental angulation, SP, and instability diagnosis rate increased after adequate analgesia was administered. A single dose of ketorolac 30 mg IM injection is a safe and convenient regimen to provide effective analgesia for dynamic radiologic examinations. This short-term pain relief facilitates the reliable functional imaging of the spine and is a feasible option for the diagnosis of intervertebral instability.

## Figures and Tables

**Figure 1 jcm-10-03984-f001:**
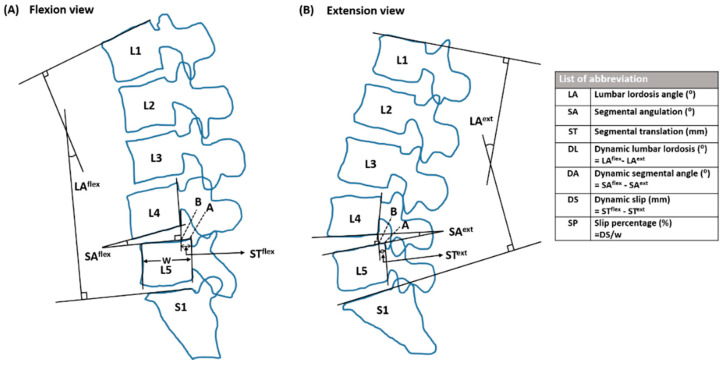
The measurement diagram for detecting motion segments of lumbar spine in flexion (**A**) and extension (**B**) views. Lumbar lordosis angle (LA) was defined as the angle between the tangential lines of the superior endplates of L1 and S1. To measure the segmental angulation (SA), tangent lines were drawn along the lower endplate of superior vertebra (such as L4) and upper endplate of inferior vertebra (such as L5); those two lines form SA. To measure segmental translation (ST), a perpendicular line from the posterior margin of lower endplate of superior vertebra of L4 to the line of upper endplate of inferior vertebra of L5 was added, and the length between A and B was defined as ST. The distance between the anterior and posterior walls of the L5 vertebra was defined as the vertebral body width (w). Dynamic lumbar lordosis (DL) was defined as the difference in lumbar lordosis angle between flexion and extension; dynamic segmental angulation (DA) was defined as the sagittal angulation change between flexion and extension; dynamic slip (DS) was the difference in segmental translation between flexion and extension. The slip percentage (SP) (%) was equivalent to DS divided by vertebral body width (w).

**Figure 2 jcm-10-03984-f002:**
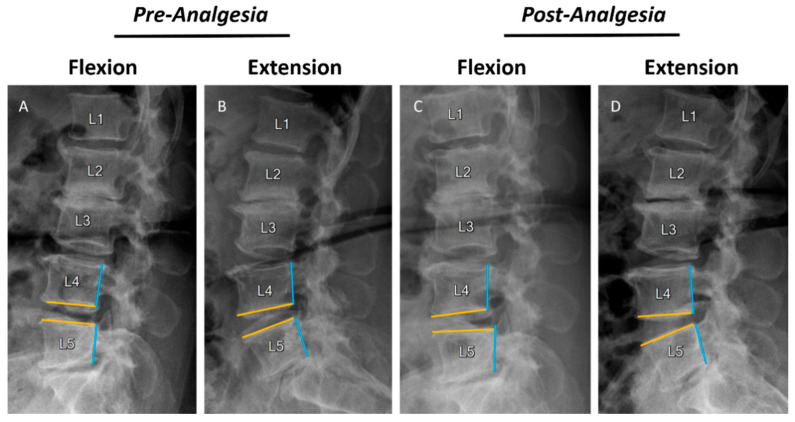
Case demonstration of radiographic change after analgesia. The radiographs showed the effect of radiographic change after intramuscular ketorolac injection in a 57-year-old patient with chronic low back pain. (**A**,**B**) X-ray image of lumbar spine flexion and extension before analgesic treatment. L45 was identified as a stable motion segment (L45, DA: 6°, DS: 0.5 mm, SP: 1.2%) with minimal displacement and angulation. (**C**,**D**) X-ray images of lumbar spine flexion and extension after analgesic treatment. Both anterior displacement and segmental angulation at L45 were increased after analgesic treatment. (L45, DA: 25.4°, DS: 4.7 mm, SP: 12%). Unstable spondylolisthesis at L45 was identified after analgesic treatment (blue line: posterior walls of vertebra; yellow line: tangent line of endplates).

**Figure 3 jcm-10-03984-f003:**
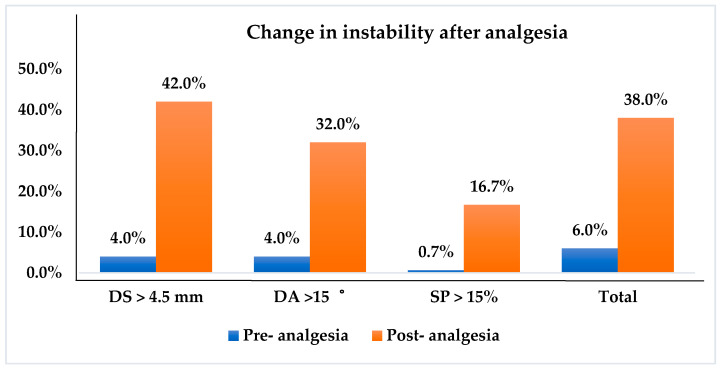
The changes in radiographic instability after analgesia. This figure shows the change in frequency in the segmental instability before and after the analgesia. There were a total of 150 motion segments. According to the diagnostic criteria for segmental instability (DS > 4.5 mm, DA > 15°, or SP > 15%), there was 4% (6), 4% (6), and 0.7% (1), respectively, of total motion segment fulfilling the diagnostic criteria. After analgesia, the diagnostic instability markedly increased to 42% (63), 32% (48), and 16.7% (25). In 100 patients, the number of potentially diagnosed cases of instability was increased from 6 (6%) before analgesia to 38 (38%) after analgesia.

**Figure 4 jcm-10-03984-f004:**
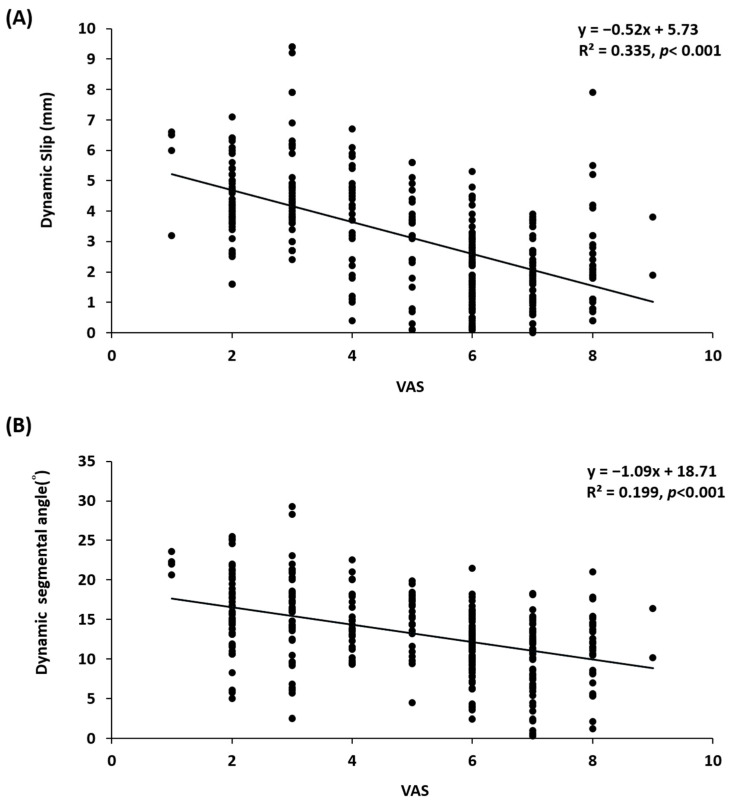
Relationships between VAS score (including all scores of baseline and final VAS) and corresponding dynamic slip (**A**), and dynamic segmental angle (**B**). The blot picture showed that both dynamic slip and dynamic segmental angle versus pain scores presented an inverse relationship.

**Table 1 jcm-10-03984-t001:** Demographics of the patients with spondylolisthesis.

N = 100	Mean	SD	Range
Age (years)	53.9 (median: 57)	11.9	28–69
Sex (F/M)	73/27		
Height (cm)	162.8	8.4	155–181
Body weight (kg)	67.6	7.9	52–87
BMI (kg/m^2^)	27.5	5.2	18.7–33.6
VAS score (mm)			
Baseline	66.9	0.97	40–90
Final	32.2	13.1	10–70
Reduction in VAS	34.7	15.5	10–70

F, female; M, man; BMI, body mass index; SD, standard deviation; VAS, visual analog scores.

**Table 2 jcm-10-03984-t002:** Parameters measured from lateral flexion/extension radiographs before and after analgesic injections.

Measured Parameters	Pre-Analgesia	Post-Analgesia	Difference ◇	Effect Size	*p*-Value
Segmental translation (mm)					
Flexion	−2.73 ± 3.49	−4.24 ± 3.94	−1.51 ± 1.34	−1.13	<0.0001
Extension	−0.79 ± 3.75	0.10 ± 4.06	0.89 ± 1.33	0.67	<0.0001
Dynamic slip (DS)	1.94 ± 1.34	4.36 ± 1.56	2.4 ± 1.36	1.76	<0.0001
Segmental angulation (°)					
Flexion	−1.36 ± 4.86	−3.64 ± 5.08	−2.27 ± 3.7	−0.61	<0.0001
Extension	9.52 ± 4.12	12.10 ± 4.46	2.57 ± 2.89	0.89	<0.0001
Dynamic segmental angle (DA)	10.89 ± 4.30	15.74 ± 4.72	4.85 ± 4.59	1.06	<0.0001
Lumbar lordosis angle (°)					
Flexion	9.06 ± 14.21	7.57 ± 12.55	−1.49 ± 8.07	−0.18	0.007
Extension	47.70 ± 11.81	50.18 ± 11.23	2.48 ± 0.55	4.51	<0.0001
Dynamic lumbar lordosis (DL)	39.03 ± 13.61	42.61 ± 12.94	3.57 ± 8.07	0.44	<0.0001
Slip percentage (SP) (%)	4.95 ± 3.41	11.11 ± 4.00	6.23 ± 3.46	1.80	<0.0001

◇, difference in measure parameters between post-analgesia and pre-analgesia status.

**Table 3 jcm-10-03984-t003:** Parameters measured from flexion/extension radiographs between moderate and severe pain groups.

	T0	Change T1–T0	*p* Value	SRM	AMD (95% CI) T1	*p* Value *
Segmental Translation (mm)						
Flexion						
M group	−2.61 ± 3.45	1.41 ± 1.37	0.41	1.03	−0.18 (−0.61, 0.26)	0.42
S group	−2.82 ± 3.53	1.60 ± 1.31		1.21		
Extension						
M group	−0.79 ± 3.73	0.90 ± 1.46	0.92	0.62	0.02 (−0.42, 0.45)	0.94
S group	−0.78± 3.79	0.88 ± 1.23		0.72		
Dynamic slip						
M group	1.82 ± 1.30	2.32 ± 1.42	0.47	1.63	−0.16 (−0.61, 0.28)	0.47
S group	2.03 ± 1.36	2.48 ± 1.33		1.87		
Slip percentage (%)						
M group	4.62 ± 3.34	6.03 ± 3.55	0.52	1.70	−0.37 (−1.50, 0.77)	0.52
S group	5.21 ± 3.46	6.40 ± 3.42		1.87		
Segmental angulation (°)						
Flexion						
M group	−1.01 ± 4.34	1.68 ± 3.15	0.07	0.53	−1.10 (−2.30, 0.11)	0.07
S group	−1.65 ± 5.24	2.76 ± 4.09		0.67		
Extension						
M group	10.27 ± 4.07 ^@^	2.30 ± 2.99	0.30	0.77	−0.51 (−1.43, 0.43)	0.29
S group	8.92 ± 4.08 ^@^	2.79 ± 2.80		0.99		
Dynamic segmental angle (°)						
M group	11.28 ± 3.93	3.98 ± 4.15	0.03	0.96	−1.60 (−3.06, −0.14)	0.03
S group	10.56 ± 4.55	5.66 ± 4.82		1.18		
Lumbar lordosis angle (°)						
Flexion						
M group	9.47 ± 11.93	−0.03 ± 7.07	0.08	−0.004	−2.81 (−6.00, 0.38)	0.08
S group	8.70 ± 15.99	2.79 ± 8.69		0.32		
Extension						
M group	49.00 ± 11.27	2.23 ± 5.55	0.68	0.40	−0.47 (−2.69, 1.74)	0.67
S group	46.58 ± 12.24	2.69 ± 5.53		0.49		
Dynamic lumbar lordosis (°)						
M group	39.52 ± 12.04	2.20 ± 6.89	0.12	0.32	−2.54 (−5.75, 0.66)	0.12
S group	38.62 ± 14.90	4.74 ± 8.85		0.54		

T0: before analgesia; T1: after analgesia; SRM: standardized response mean; M: moderate pain group; S: severe pain group; AMD: adjusted mean difference between M and S groups using general linear model {adjusting for age}; *, a *p* value after adjusting for age; ^@^, the S group had significantly lesser extension angle than the M group at pre-analgesia status (*p* = 0.046). Data are presented as mean ± SD, unless otherwise stated.

## Data Availability

The data presented in this study are available upon request from the corresponding author. The data are not publicly available due to privacy restrictions.

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
