# Peer review of "Pain Control Affects the Radiographic Diagnosis of Segmental Instability in Patients with Degenerative Lumbar Spondylolisthesis"

_jcm, 2021, doi:10.3390/jcm10173984_

Round 1
Reviewer 1 Report
I believe this is a well-written article containing informative interesting issues. However, some points need to be pointed out for revision before acceptance.
First of all, I think the authors should amend their title. The current article is limited to “degenerative Spondylolistheses (SLTs). Therefore, I belive this must appear clearly in the title as well.
Methods
- Inclusion criteria ; low grades of DLS – please state this indication with more detail. By which classification or criteria did you define “low grade” DLS.
- Why did you have to include those allergic to NSAIDs? Not just those allergic to Ketorolac.
- *important ; at which time point did you perform your post-NSAIDs injection radiograph. The vas score seems to be collected 30mins after the injection, and was the time for radiograph the same? Additionally, what was your rationale for this specific time point. Do you have reference that this time point is of maximum drug effect after IM injections?
- *critical ; as a prospective study, it would have been a very much good quality comparative study if the authors had a comparative group with placebo IM injections. It would be good if you could consider this point and add some comments about this – why you couldn’t conduct this study in that form, or if you have any further plans for any possible further studies.
- Subgroup analysis ; moderate vs severe groups – are not described in the Methods.
Results
- Figure 3. Which point of VAS score is presented in this graph. While the authors state within section 3.5 that both basline and final VAS scores show inversed relationship, it is not clear which VAS score is presented in this graph. It seems like the pre-injection VAS, then what are your correlation in case of final VAS.
- It would be greatly appreciated if you could present a representative case image, showing the significant difference of motion measured on radiographs before and after NSAIDs injection.
Discussion
- I feel that the discussion is relatively very well written.
I do not find any critical weak points in this article.
I believe it will be a good additive knowledge to spinal specialists after proper amendment.
Reviewer 2 Report
"Pain Control Affect the Radiographic Diagnosis of Segmental Instability in Patients with Spondylolisthesis" is a well written paper and could be of potential interest in literature.
As well reported, Lilius et al. and Williams et al., did not detect a change in flexion extension ROM or gain in lumbar curvature after pain relief.
Even if it's an old paper, Lilius stated that 36% pt reported persisting benefit at the three month follow-up, independent of the mode of treatment given. He concluded that facet joint injection is a non-specific method of treatment and the good results depend on a tendency to spontaneous regression and to the psychosocial aspects of back pain" .
In the present paper, despite the excellent introduction, the measure of the VAS was recorder 30 min after pain relier administration and no other measures were reported.
Moreover, there wasn't a placebo group (such as saline solution) to compare the results.
Considering the potential psychosocial aspects of back pain patients, a second control once the analgesia has worn off and after 3-6 months should be administered.
Round 2
Reviewer 2 Report
Accepted after revisions.